# Undetected Causes of Death in Hospitalized Elderly with COVID-19: Lessons from Autopsy

**DOI:** 10.3390/jcm10071337

**Published:** 2021-03-24

**Authors:** Astrid Malézieux-Picard, Cecilia Ferrer Soler, David De Macedo Ferreira, Emilie Gaud-Luethi, Christine Serratrice, Aline Mendes, Dina Zekry, Gabriel Gold, Johannes Alexander Lobrinus, Grégoire Arnoux, Fulvia Serra, Virginie Prendki

**Affiliations:** 1Division of Internal Medicine for the Aged, Department of Rehabilitation and Geriatrics, University Hospitals of Geneva, Hôpital des Trois-Chêne, 1226 Thônex-Genève, Switzerland; davidjose.demacedoferreira@hcuge.ch (D.D.M.F.); christine.serratrice@hcuge.ch (C.S.); dina.zekry@hcuge.ch (D.Z.); virginie.prendki@hcuge.ch (V.P.); 2Division of Geriatrics, Department of Rehabilitation and Geriatrics, University Hospitals of Geneva, Hôpital des Trois-Chêne, 1226 Thônex-Genève, Switzerland; Cecilia.ferrersoler@hcuge.ch (C.F.S.); Emilie.luethi@hcuge.ch (E.G.-L.); aline.mendes@hcuge.ch (A.M.); Gabriel.gold@hcuge.ch (G.G.); 3Division of Pathology, University Hospitals of Geneva, 1205 Geneva, Switzerland; Johannes.a.lobrinus@hcuge.ch (J.A.L.); gregoire.arnoux@unige.ch (G.A.); fulvia.serra@hcuge.ch (F.S.); 4Division of Infectious Disease, University Hospitals of Geneva, 1205 Geneva, Switzerland

**Keywords:** elderly patients, COVID-19, autopsy

## Abstract

Background: Mechanisms and causes of death in older patients with SARS-CoV-2 infection are still poorly understood. Methods: We conducted in a retrospective monocentric study, a clinical chart review and post-mortem examination of patients aged 75 years and older hospitalized in acute care and positive for SARS-CoV-2. Full body autopsy and correlation with clinical findings and suspected causes of death were done. Results: Autopsies were performed in 12 patients (median age 85 years; median of 4 comorbidities, mainly hypertension and cardiovascular disease). All cases showed exudative or proliferative phases of alveolar damage and/or a pattern of organizing pneumonia. Causes of death were concordant in 6 cases (50%), and undetected diagnoses were found in 6. Five patients died from hypoxemic respiratory failure due to coronavirus disease 2019 (COVID-19), five had another associated diagnosis and two died from alternative causes. Deaths that occurred in the second week were related to SARS-CoV-2 pneumonia whereas those occurring earlier were related mainly to heart failure and those occurring later to complications. Conclusions: Although COVID-19 hypoxemic respiratory failure was the most common cause of death, post-mortem pathological examination revealed that acute decompensation from chronic comorbidities during the first week of COVID-19 and complications in the third week contributed to mortality.

## 1. Introduction

The new coronavirus, Severe Acute Respiratory Syndrome Coronavirus-2 (SARS-CoV-2), has caused to date close to 2 million deaths worldwide. Older patients, who present the highest prevalence of multiple chronic diseases, are the hardest hit among the population and advanced age is the strongest predictor of mortality [1,2,3]. Although several autopsy studies of coronavirus disease 2019 (COVID-19) have already been published, most include only individual or small autopsy case studies with limited post-mortem examinations [4,5,6]. Complete autopsies of very old patients are rare [7,8,9]. These studies suggest that inflammation, hypercoagulation and endothelitis are the main pathophysiological processes leading to death in COVID-19. However, the mechanisms and causes of death in older patients with SARS-CoV-2 infection are still poorly understood [10,11,12]. Such information may help clinicians to improve clinical management and to reduce mortality [13,14,15,16]. In order to explore this issue, we correlated clinical findings with pathological findings and analyzed the causes of death in a case series of 12 older patients who had been hospitalized in acute geriatric care.

## 2. Materials and Methods

### 2.1. Design, Setting and Population Study

The study was set up in the geriatrics hospital in Geneva, Switzerland, with 176 beds in acute care units dedicated to infected SARS-CoV-2 patients. These patients were ineligible for intensive care according to goals of care determinations. The hospital serves a population of about 500,000 inhabitants. We included all patients hospitalized in the geriatrics hospital with a positive PCR SARS-CoV-2 test from 13 March to 2 May 2020, who died and for whom we obtained a written autopsy authorization from their representative. All patients were part of the COVIDAge study, a retrospective monocentric study approved by the ethical committee of Geneva and registered in clinicaltrials.gov [17].

### 2.2. Data Collection

Clinical, biological, and radiological data were collected from electronic medical records, including causes of death determined by clinicians. All patients were evaluated within the first 24 h using the following clinical and geriatric assessment scales that are validated for use in older populations: Cumulative Illness Rating Scale-Geriatric (CIRS-G, range: 0–56), Functional Independence Measure (FIM, range: 18–126), Clinical Frailty Scale (CFS, range: 1–9), Body Mass Index (BMI), Pneumonia Severity Index (PSI, range: 51–395), CURB-65 Severity Score (CURB-65, range: 0–5) [18,19,20,21,22,23] (Appendix A).

### 2.3. Autopsy Techniques

All autopsies were performed with a post-mortem delay between 25 and 96 h (mean 45 h) and conducted by two resident pathologists at the Division of Pathology, (FS and GA). In all cases a complete full body autopsy was performed, including brain in all but one case. All organs where eviscerated and analyzed either immediately or after 48 h of 4% formalin fixation. For each patient, 4 kidney samples were taken for electron microscopy analysis and in most cases multiples frozen sections for further analysis. After fixation, organs were dissected and multiple tissue samples taken and paraffin-embedded. 3–4 µm thick sections were prepared and stained with hematoxylin and eosin. Special stains and/or immunostainings were performed on selected samples. All slides were analyzed by the senior pathologist (JAL) and the same two residents (FS, GA), aware of the clinical history and evolution of the patient.

### 2.4. Clinical and Pathological Confrontation

Causes of death reported by clinicians in the record and the autopsy request were compared with autopsy findings.

## 3. Results

### 3.1. Clinical Characteristics

During the study period, 264 older people were hospitalized with COVID-19. The youngest patient was 75 years old and the oldest 95 years old (median of 86 years old). Post-mortem examinations were conducted on 12 (14.8%) of the 81 patients with SARS-CoV-2 who died in our hospital. Patients’ characteristics are illustrated in Table 1.

The median age was 85 years old (interquartile range (IQR), 75–95 years); half were female. Eight patients (75%) were living at home, three patients had a prior hospitalization in the preceding 6 months. The most frequent comorbidities were hypertension (83%), heart disease (50%), chronic kidney disease (50%), dementia (42%), and diabetes (33%). Only one patient had known chronic obstructive pulmonary disease. The median and IQR of FIM, CFS and CIRS-G was 86 (25–107), 8 (5–9), and 22 (11–32) respectively, consistent with moderately high levels of functional impairment, frailty, and comorbidity burden. The mean number of medications at admission was 9 (3–18) and the mean BMI was 25.6 (21.4–47.7; 3 missing data). The average duration from first symptoms to death was 9 days (3–25). The characteristics of autopsied patients with non survivors in the COVIDAge study were quite similar [17]. The median age of the non-survivors was 87 years (IQR: 80.5–93.3), the median CIRS-G was 21.2 (IQR: 15.9–26.5), and 69.3% received antibiotics. There were slightly more male deaths (63.2%) in the COVIDAge cohort and the CFS was lower 6.5 (IQR: 5.1–8.9) but not statistically significant.

The most common symptoms at admission were fever (100%), cough (83%), asthenia (77%) and dyspnea (72%). Three patients had delirium and one had digestive symptoms. CURB-65 and PSI scores were 2 (1–3) and 137 (106–185) respectively. Pulmonary infiltrates on chest X-ray were multifocal in ten cases and local in two. Thoracic CT scan was performed in two patients and revealed multifocal infiltrates and pleural effusion.

During hospitalization, all patients required a fraction of inspired oxygen (FiO_2_) greater than 50% and had fever. All patients developed hypoxemic respiratory failure, 5 (42%) acute heart failure treated with diuretics, 4 (33%) acute renal failure, and one delirium. An associated bacterial pneumonia was suspected by the clinician in 8 (66%) patients and antibiotic therapy was prescribed despite the absence of bacterial respiratory pathogens in sputum and blood cultures. One patient who had *Escherichia coli* bacteremia developed pancytopenia. All patients were treated with prophylactic anticoagulation therapy except one who initially received higher doses for the treatment of deep vein thrombosis. Patient 3 received hydroxychloroquine and lopinavir-ritonavir according to local guidelines.

Patients 2, 3, 4, and 9 died in the first week of disease. Patient 3 died suddenly a few hours after admission, a rhythm disorder was suspected. Patients 4 and 9 died with signs of cardiac failure and suspicion of associated bacterial pneumonia. These three last patients had severe cardiac comorbidities. Five patients (42%) died in the second week of disease. They developed acute respiratory distress syndrome (ARDS) and in three cases acute renal failure. Patients 6, 10, and 11 died after 15 days. Patient 6 had been hospitalized one month before for a fall, leading to a deep vein thrombosis diagnosis. He developed a severe delirium due to hypernatremia, nosocomial bacterial pneumonia with left heart failure and a severe macroscopic hematuria leading to discontinuation of the anticoagulation. Patient 10 presented with multiorgan failure at day 20. Patient 11 had persistent fever and hypoxemia, developed acute renal failure with liver cholestasis, and increasing oxygen needs at day 15; he was treated with 3 antibiotics successively.

### 3.2. Autopsy Findings

#### 3.2.1. Lungs

At gross examination, lungs of all patients were heavy and congested, with a mean weight of 1584 g (range 1050–1984). At section, they showed condensation in patchy areas, sometimes even fibrous thick areas, or complete consolidation of the entire parenchyma. An interesting characteristic was the clear limit between involved and normal appearing areas of lung tissue. This pattern is not typically seen in ARDS cases and fits well with the radiologic aspect. In cases of bacterial superinfection, patchy round white dense areas were observed. Upon histological examination, alveolar damage (AD) at the exudative or proliferative phase and/or organizing pneumonia (OP) were observed and could be present alone or together. Lung damage was severe in 8 cases (67%). Of these, the pattern of exudative phase of AD was found prevalent in 6 cases (associated with a short time duration of illness) and proliferative phase of AD and OP in 2 cases (associated with a more prolonged illness). In four cases, signs of bacterial superinfection pneumonia were present (3 out of 4 with bronchoaspiration signs such as the presence of food particles). Microthrombi were present in three cases, of which one had central pulmonary embolism in the left pulmonary artery (patient 6). One case showed eosinophilic pneumonia (patient 11). Detailed lung pathology is shown in Table 2. An illustration of a typical SARS-CoV-2 lung injury is show in Figure 1.

#### 3.2.2. Heart and Vessels

All twelve patients presented an ischemic and/or hypertensive cardiopathy, with signs of heart failure in four cases. Coronary stenosis or stenting were seen in eleven patients, and old or recent myocardial damage in nine patients. Moderate to severe generalized atherosclerosis was found in all patients (50% moderate and 50% severe). No myocarditis or vasculitis was discovered.

#### 3.2.3. Kidney

Chronic vascular nephropathy was observed in all cases, signs of diabetic nephropathy were found in two and acute or subacute tubular necrosis in five patients. In one case, predominantly distributed in the superficial renal medullary, nonspecific foci of acute tubulointerstitial nephritis were present. Patient 10 had a thrombus in one interlobular artery. No structures typical of coronavirus particles were seen in electron microscopy.

#### 3.2.4. Brain

A brain autopsy was performed in eleven patients. Neurodegenerative signs were present in four patients (Alzheimer and/or Lewy body disease). Ten patients presented a variable degree of atherosclerosis of the circle of Willis. Patient 10 was the only one with microthrombi of a few small meningeal vessels and acute cortical microinfarcts of the frontal lobes. Patient 12 had one subacute infarct of the left pons. Patient 8 had an old bilateral parasagittal infarct centered on the precentral and postcentral gyri. No encephalitis or vasculitis was observed.

#### 3.2.5. Main Other Findings

Patients 1, 7, and 11 had a cirrhosis and/or liver steatosis, patient 10 had metastatic urothelial carcinoma, patient 6 had a necrotizing myopathy and patient 5 cachexia. Bone marrow was normal or reactive, without hematologic disease.

### 3.3. Clinicians’ Cause of Death and Pathologists’ Findings

The clinicopathological comparison of causes of death was concordant in six cases (50%) (Table 3). Undetected diagnoses (new primary or contributory cause of death) were found in six cases: Pulmonary embolism (one case), acute myocardial infarct (one case), eosinophilic pneumonia (one case) and heart failure (three cases). The cause of death presumed by the clinician was in each case a hypoxemic respiratory failure due to COVID-19. At autopsy, severe hypoxemic pneumonia (ARDS) due to COVID-19 was confirmed as the unique cause of death in only five cases. Five patients had additional diseases: autopsies of patients 2, 3, and 4 revealed signs of acute heart failure (pulmonary edema, shock liver and multiple organ congestion), patient 10 had a small acute cardiac infarct and patient 11 had signs of eosinophilic pneumonia. Autopsy revealed that two patients died from another cause than COVID-19: Patient 6 from a pulmonary embolism (anticoagulation had been discontinued due to severe hemorrhage) and bacterial pneumonia (21 days after symptom onset) and patient 9 from a probable arrhythmia due to pulmonary hypertension and ischemic cardiac disease.

Interestingly, deaths that occurred in the second week of active disease were related only to SARS-CoV-2 pneumonia whereas deaths that occurred earlier were related, at least in part, to cardiac comorbidities, mainly heart failure; deaths that occurred later were related to complications of the disease and/or its therapy (Figure 2).

Eight patients had been treated with antibiotics for bacterial pneumonia which was not confirmed at autopsy in five cases (patients 4, 8, 9, 10, 11). One patient died rapidly with a bacterial pneumonia without antibiotics (patient 2). Five patients developed acute tubular necrosis but this was not notified by the clinician on the autopsy questionnaire for three of them.

## 4. Discussion

To our knowledge, this is the first autopsy series of very old patients hospitalized in a geriatrics hospital and who died with a SARS-CoV-2 confirmed infection. The overall comorbidity burden was high and associated with severe frailty. The autopsies identified different or additional contributory causes of death and potentially treatable undetected diagnoses in approximately half the cases. This is higher than a similar report in a slightly younger German cohort (mean age 79 years) where SARS-CoV-2 pneumonia was not the only cause of death in 11% of the cases and was not the cause at all in 5% [24].

An interesting finding of our study is the timing of the different causes of death (Figure 2) suggesting that early detection and therapy of decompensated comorbidities, especially heart failure, may help decrease mortality during the first week of the disease in older patients and that prevention of late complications including ischemic and thromboembolic events and side effects of possibly unnecessary antibiotic therapy may improve survival after the second week.

Bacterial superinfection was shown in half of the cases on antibiotics. The current literature has highlighted that bacterial pneumonia was rarely associated with COVID-19 infection whereas patients were often treated with antibiotics, as suggested in several therapeutic guidelines generated at the beginning of the pandemic. In fact, co-infections are rare and have been mainly described in intensive care units due to prolonged intubation [25]. Our findings support the principle of limiting routine prescription of antibiotics in SARS-CoV-2 pneumonia and extend it to the older population above 75 years of age [26]. However, the diagnostic of pneumonia in elderly patients is challenging because symptoms are less specific and decompensated comorbidities are frequently associated [27]. The differentiation between viral versus bacterial pneumonia may also be difficult as the microorganism causing bacterial pneumonia is identified in only a minority of patients [28]. Patient 2 did not receive antibiotic but signs of bacterial pneumonia were finally found on the autopsy.

We report one case of eosinophilic pneumonia, a rare and heterogeneous syndrome. Its clinical presentation may include fever, dry cough, dyspnea, chest pain, and bilateral reticular ground glass opacities on imaging [29]. In our case, we believe the use of piperacillin-tazobactam was the most probable cause even though, to our knowledge, only 4 cases of eosinophilic pneumonia due to piperacillin-tazobactam have been described in the literature [30]. Importantly, prior cases of SARS-CoV-2 related eosinophilic pneumonia have been reported, suggesting a potential link between SARS-CoV-2 and eosinophilic pneumonia; further autopsy studies will be needed to clarify this issue [31,32,33].

Other pulmonary findings of our cohort were similar to those described in other studies [34,35,36]. We observed the two main pulmonary patterns found in common cases of viral pneumonia: Alveolar damage and organizing pneumonia. As expected, the exudative phase of alveolar damage was the prevalent pattern in patients who died early (but not only) and the proliferative phase of diffuse alveolar damage or the organizing pneumonia, were prevalent in patients who died at least 6 days after the onset of the symptoms. The presence of pulmonary microthrombi has been extensively discussed in the medical literature, mainly in relation to potential preventive therapy [37]. Wichmann et al. showed that 58% of deaths during COVID-19 were caused by venous thromboembolism [9]. Rapkiewicz et al. and Lax et al. showed microthrombi in all autopsies [8,38]. On the other hand, Bradley et al. observed rare microthrombi and no endothelitis in a cohort of 14 patients [39]. In our series, only three patients had fibrinous microthrombi including one with a history of venous thrombosis in whom longstanding anticoagulation therapy had been discontinued because of severe hematuria.

Many of our patients had acute heart failure and one had an acute myocardial ischemia but unlike other studies, no myocarditis was found [40].

Acute renal failure was also common and related to decompensated chronic vascular and diabetic nephropathies. The incidence of acute kidney injury (AKI) during SARS-CoV-2 infection varies from 0.5% to 80.3% [41]. It is more frequent in critically ill patients. Different mechanisms have been described: Direct effect of the virus and secondary mechanisms linked to the hemodynamic (hypo-perfusion), thrombotic micro-angiopathy, humoral response to the virus, and activation of the complement system [41]. Histologically, nonspecific acute or subacute tubular necrosis was observed in five of our patients and none of them had lymphocyte infiltration. This is in line with the results of Pei et al. who described varying degrees of acute tubular necrosis without lymphocyte infiltration [42]. We also observed a thrombus in one interlobular renal artery and nonspecific foci of acute tubulointerstitial nephritis in another one, but no evidence of endarteritis, nor tubulitis or fibrinous microthrombi as mentioned in other studies. [43]. Contrary to several other reports, electron microscopy did not reveal any coronavirus particle in our series [44].

The post-mortem brain analysis showed common lesions for geriatric patients without any pathology directly related to SARS-CoV-2 infection; this is consistent with other findings in older patients [45].

We found a necrotizing myopathy in one case probably related to treatment with statins for many years. Among older patients, necrotizing myopathies are most frequently triggered by statins and we did not find any description of SARS-CoV-2 induced necrotizing myopathy in the literature [46].

Our study has several limitations. PCR was not performed in the tissue to detect the virus, since the infection was already confirmed. As well, given its monocentric retrospective design and its relatively small sample size it may not be representative of all older populations. However, it includes all autopsied COVID-19 cases during the first wave in a well described cohort of older people from the COVIDAge study representing a population with a broad range of comorbidities and frailty status often encountered in acute geriatric care [17]. Importantly, there were no major demographic or clinical differences between autopsied and non-autopsied deceased patients.

## 5. Conclusions

The post-mortem pathological examination revealed one or several undetected diagnoses in half the cases. Although COVID-19 hypoxemic respiratory failure was the main cause of death, autopsies showed that acute decompensation from chronic comorbidities during the first week of COVID-19 disease and complications in the third week were either a direct cause of death or contributed significantly to mortality in older hospitalized COVID-19 patients. Early detection and treatment of heart failure, prevention of thromboembolic complications, and avoidance of unnecessary antibiotic therapy may help decrease mortality in older patients hospitalized with COVID-19. More studies based on autopsies are needed in very elderly patients.

## Figures and Tables

**Figure 1 jcm-10-01337-f001:**
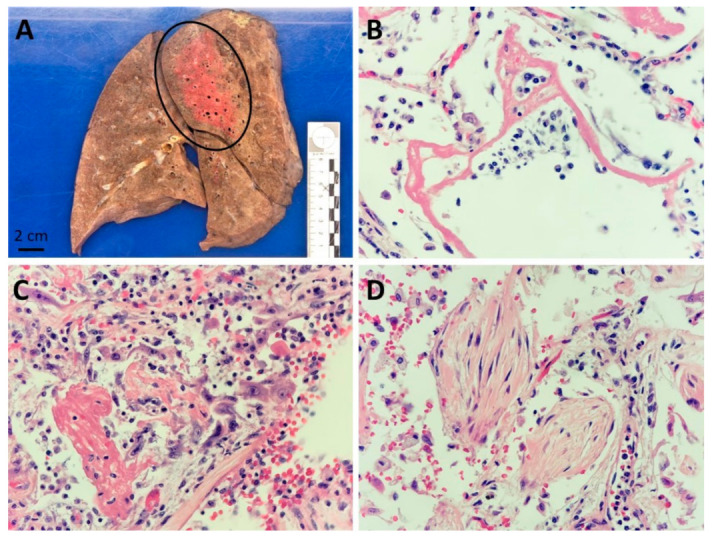
Lung pathology associated with SARS-CoV-2 infection: (**A**) Sagittal macroscopic section of a formalin fixed right lung, showing a condensed area in the posterior part of the upper lobe (encircled area), patient n 7; (**B**) Alveolar damage at exudative stage with prominent eosinophilic hyaline membranes (HE 400×), patient n 2; (**C**) Alveolar damage at proliferative stage with proliferation of alveolar pneumocytes (HE 400×), patient n 10; (**D**) Organizing pneumonia with intra-alveolar fibroblast plugs (HE 400×), patient n° 11.

**Figure 2 jcm-10-01337-f002:**
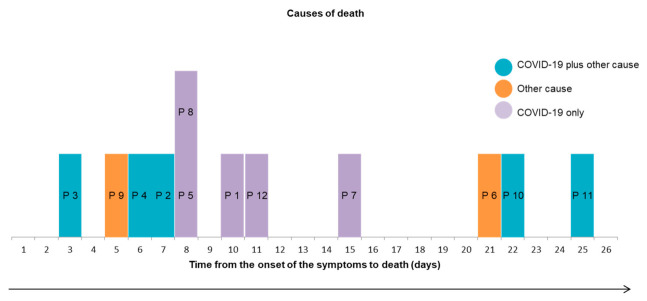
Causes of death according to time to death. Abbrevations: P: Patient.

**Table 1 jcm-10-01337-t001:** Clinical characteristics of the 12 patients.

Patient No.	Sex	Age (years)	Clinical Medical History	Time from Symptoms to Death (Days)	CIRS-G (Range 0–56)	Clinical Frailty Scale (Range 1–9)	FIM (Range 18–126)	Radiologic Pulmonary Infiltrates
1	F	87	Arterial hypertension, primary biliary cirrhosis, chronic lymphocytic leukemia	10	27	9	107	multifocal
2	F	86	Asthma, Factor V Leiden, lower limb neuropathy	7	19	8	MD	multifocal
3	F	83	Arterial hypertension, ischemic and valvular heart disease, arteriosclerosis, stroke, dementia, diabetes, lower limb neuropathy	3	23	6	MD	multifocal
4	M	86	Dilated cardiomyopathy, pacemaker	6	15	5	101	multifocal
5	F	95	Arterial hypertension, dementia, diabetes	8	24	9	MD	multifocal
6	M	91	Arterial hypertension, ischemic heart disease, lower limb arterial insufficiency, dyslipidemia, asthma, deep vein thrombosis	21	11	5	86	local
7	F	81	Arterial hypertension, dementia, chronic hepatitis C, chronic kidney failure, breast cancer, depression	15	22	8	25	multifocal
8	F	88	Arterial hypertension, pulmonary embolism, epilepsy, chronic kidney failure	8	24	8	MD	local
9	M	88	Arterial hypertension, ischemic, valvular and rhythmic heart disease, COPD, pulmonary arterial hypertension, chronic kidney failure	5	21	5	MD	multifocal
10	M	81	Arterial hypertension, ischemic heart disease, diabetes, dyslipidemia, chronic kidney failure, metastatic bladder cancer	22	21	8	103	multifocal
11	M	75	Arterial hypertension, ischemic heart disease, arteriosclerosis, IgA nephropathy, chronic kidney failure, stroke, dementia, cirrhosis, arteriosclerosis	25	32	6	73	multifocal
12	M	81	Arterial hypertension, diabetes, dyslipidemia, stroke, dementia, chronic kidney failure, lower limb neuropathy	11	13	MD	35	multifocal

Abbreviations: CIRS-G: Cumulative Illness Rating Scale-Geriatric; COPD: chronic obstructive pulmonary disease; FIM: Functional Independence Measure; MD: Missing Data.

**Table 2 jcm-10-01337-t002:** Autopsy finding.

Patient No.	Time from Symptoms to Death (Days)	Lung Autopsy Findings								Other Non Pulmonary Death-Related Autopsy Findings
		AD: Patchy Exudative	AD: Diffuse Exudative	AD: Patchy Proliferative	AD: Diffuse Prolferative	OP: Patchy	OP: Diffuse	Bacterial Pneumonia	Other Lung Findings	
1	10		XXX							
2	7		XXX					Foci		Signs of heart failure
3	3	XX								Signs of heart failure
4	6		XXX	XX						Signs of heart failure
5	8		XXX			X		Foci		
6	21					X		Large	Central left pulmonary embolism	
7	15		XX							
8	8	X					XX		Interstitial pneumonia Rare peripheric thrombi	
9	5	X		X					Signs of pulmonary hypertension	Ischemic heart disease
10	22	X			XXX				Peripheric thrombi and microthrombi	Small acute septal myocardial infarct and signs of heart failure
11	25	XX					XXX		Eosinophilic pneumonia	
12	11		XXX			X		Foci		

Abbreviations: AD: Alveolar Damage, OP: Organizing Pneumonia, X: Slight, XX: Moderate, XXX: Severe.

**Table 3 jcm-10-01337-t003:** Clinical and pathological confrontation.

Patient No.	Sex	Age (Years)	Time from Symptoms to Death (Days)	Complications during Hospitalization	Clinical Suspicion Cause of Death	Cause of Death at Autopsy *
1	F	87	10	Anemia due to bleeding; Acute renal failure	HRFdue to COVID-19	HRFdue to COVID-19
2	F	86	7		HRFdue to COVID-19	HRFdue to COVID-19 and heart failure
3	F	83	3	Urinary retention	HRFdue to COVID-19	HRFdue to COVID-19 and heart failure
4	M	86	6	Global cardiac failure; Sacral ulcer; Suspicion of bacterial pneumonia;	HRFdue to COVID-19 and heart failure	HRFdue to COVID-19 and heart failure
5	F	95	8	Escherichia coli bacteremia; Pancytopenia; Suspicion of bacterial pneumonia; Acute renal failure	HRFdue to COVID-19 and sepsis with Escherichia coli bacteremia	HRFdue to COVID-19
6	M	91	21	Left heart failure; Delirium; Hypernatremia; Macroscopic hematuria; Suspicion bacterial pneumonia	HRFdue to COVID-19 and delirium	Pulmonary embolism and bacterial pneumonia
7	F	81	15		HRFdue to COVID-19	HRFdue to COVID-19
8	F	88	8	Arterial hypertention	HRFdue to COVID-19	HRFdue to COVID-19
9	M	88	5	Left heart failure; Suspicion of bacterial pneumonia	HRFdue to COVID-19	Probable arythmia due to pulmonary hypertension and ischemic heart disease
10	M	81	22	Atrial fibrillation; Suspicion of bacterial pneumonia; Multiorganic failure	HRFdue to COVID-19	HRFdue to COVID-19 and small acute myocardial infarct
11	M	75	25	Acute renal failure; Liver cholestase; Anemia; Suspicion of bacterial pneumonia; Unexplicated persisting inflammation	HRFdue to COVID-19	HRFdue to COVID-19 and eosinophilic pneumonia
12	M	81	11	**Hypoglycemia; Acute renal failure; Left heart failure; Suspicion of bacterial pneumonia**	HRFdue to COVID-19	HRFdue to COVID-19

Abbreviations: AD: Alveolar Damage; OP: Organizing Pneumonia; HRF: Hypoxemic Respiratory Failure. * Cause of death presumed by the pathologist. All the patients presented signs of acute respiratory distress syndrome due to COVID-19.

## Data Availability

The data presented in this study are available on request from the corresponding author. The data are not publicly available due to ethics and general data production regulation.

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
