# Peer review of "Undetected Causes of Death in Hospitalized Elderly with COVID-19: Lessons from Autopsy"

_jcm, 2021, doi:10.3390/jcm10071337_

Round 1

Reviewer 1 Report

Dear Authors,

first of all, I thank You for giving me the opportunity to read this Your  manuscript.

I read it with great interest and attention.

You are right. More than one drug practice has no justification in older patients with SARS-CoV-2, and Your study confirmed it.

I have only minor comments:

1) Table 3 was unclear. Are there typographical mistakes ? Please, make this table clearer.

2) References shoud be re-written in line with journal instructions.

3) Acute or subacute tubular necrosis was found in five patients (see line 177). Please, discuss these findings. 

4) In Your study,  eight patients had been treated with antibiotics for bacterial pneumonia which was not confirmed at autopsy in five case. On the other hand, one patient died with a bacterial pneumonia without antibiotuics (please, see lines 220-222). Discussion on the elements of suspicion for a bacterial pneumonia would be very useful for the reader. Please, add this in Your discussion section.  

Reviewer 2 Report

The authors present a nice summary of 12 autopsies of geriatric COVID-19 patients who did not have access to the intensive care unit. An interesting aspect of the study is the correlation between disease duration and causes of death. This is a descriptive-retrospective study with the appropriate limitations that come with this study design.

I have only minor comments:

  1. Introduction: it is no longer entirely true that there are very few studies with full autopsies. Already in mid-2010 some larger series were available (compare e.g. overview in reference 14). This should be acknowledged.
    While it is true that these studies did not specifically look at very old patients, these of course formed the bulk of the deceased studied in all autopsy studies.
  2. The authors should specify how the autopsies were obtained or why many autopsies were not obtained. Was the consent of the relatives required? Were there official orders? In any case, with an autopsy rate of 14.8%, there is reason to fear that there may be a selection bias.
  3. Indeed, it is not essential to repeat SRAS-CoV-2 PCR for clinically confirmed tests at autopsy. Nevertheless, it seems important to me to emphasize that all patients had a positive PCR test during their lifetime and when the test was done in relation to disease progression. Perhaps there is a way to incorporate this information into Fig. 2.
  4. Figure 1a is not entirely convincing. It is fixed lung tissue that has been photographed without scale. There is an impression that the lower part is honeycombed-emphysematous, while the upper part appears consolidated. Is this a late stage? The assignment to the respective patient should be made clear for all images.
